# Impact of Sars-Cov-2 on access to healthcare by person with disabilities

**Nkosi Nkosi Botha** [1,2]*, **Fortune Selase Atsu**[1], **Edward Wilson Ansah**[1], **Cynthia Esinam Segbedzi**[1], **Sarah Annim**[1]

**1** Department of Health, Physical Education and Recreation (HPER), University of Cape Coast, Cape Coast, Ghana, **2** Air Force Medical Centre, Air Force Base, Takoradi, Ghana

* saintbotha2015@gmail.com

## Abstract

The Sars-Cov-2 pandemic has ravaged societies at their very core and deepened pre-existing inequalities. Meanwhile, persons with disabilities (PwDs), the most oppressed group in Ghana that live in poor and deplorable conditions are most like to be negatively impacted by the Sars-Cov-2 crisis. Therefore, the aim of this study is to explore how the Sars-Cov-2 pandemic is influencing access to healthcare by PwDs in the Sekondi-Takoradi Metropolis (STM). We collected data from 17 participants, nine from the Ghana Blind Union (GBU), five from Ghana Society for the Physically Challenged (GSPC), and three from the Ghana National Association of the Deaf (GNAD). An interview guide containing 25 items was used to gather data from the participants and we employed Phenomenological Analysis (PA) approach in making sense of the data. PWDs encounter many different barriers like; i) stigma and discrimination, ii) cost and availability of transport, iii) poor attitude of healthcare staff, iv) poor communication, v) hospital environment and equipment, vi) handwashing and sanitizing facilities, vii) unsuitable washrooms, viii) cost of healthcare, ix) registration and renewal of NHIS cards, and x) loss of income as they attempt to seek healthcare during this Covid-19 era in the STM. Covid-19 pandemic has widened the disproportionate and inequality gaps against PWDs in the STM when they attempt to seek healthcare. in the face of this, STM may lead Ghana to lag in achieving the Sustainable Development Goal (SDG) 3.8, which entreats nations to provide quality healthcare for all persons including PWDs. PWDs need education and empowerment to enable them demand for their rights when accessing healthcare. The findings highlight existing gaps in the implementation of the disability law by healthcare facilities in STM and, re-focus the attention of hospital managers in STM to the healthcare needs of PWDs in STM.

**Data Availability Statement:** All necessary data is included in the work.

**Funding:** The authors received no specific funding for this work.

## Introduction

The Sars-Cov-2 pandemic has impacted society at the very core and deepened pre-existing inequalities [1]. Even during normal situations, one billion persons with disabilities (PwDs) globally are not likely to access healthcare. This situation is additionally worsened against persons living in humanitarian and precarious contexts such as the situation created by the Sars-

**Competing interests:** The authors have declared that no competing interests exist.

Cov-2 pandemic [2, 3]. Evidence [4] suggests that one billion people, or 15% of the world's population experience some form of disability, of which one-fifth or between 110 million and 190 million people experience significant disabilities. Unfortunately, PwDs are more likely to experience adverse socioeconomic outcomes than others with even less education, poorer health outcomes, lower levels of employment, and higher poverty rates [4, 5].

As the unprecedented Sars-Cov-2 outbreak created global crisis and ravaged the life and work of the general population [6], it further disproportionately impacted PwDs because of the convoluted pre-existing conditions. The interruption of vital services and support, pre-existing health ailments in some cases have left PwDs in danger of acquiring grave health ailments [1, 7]. This category of persons are also most likely to live in poverty, endure increased levels of violence, neglect and abuse, inequalities which Sars-Cov-2 has further intensified [8]. To aggravate matters, an estimated 46% of the elderly, 60 years and above live with a form of disability [6].

The plights of PwDs are vast and deep, for example, PwDs have often been sidelined from health information, and mainstream health provision, including access to goods and services [9]. The Sars-Cov-2 pandemic has thus, exposed these underlying inequalities and exacerbated the vulnerability [3, 9]. For PwDs, everyday restrictions like physical accessibility, barriers to rolling out basic hygiene protocols, access to vaccines, affordability of healthcare, limitations on health insurance, and prejudiced laws and stigma can independently or in combination be life frightening in the mid of this global pandemic [4, 10].

The situation in Africa, where disability prevalence is higher, remains largely frightening as the Sars-Cov-2 pandemic rolled back progress chalked towards sustainable growth and development [9]. Long before the Sars-Cov-2 crisis, most African countries were less likely to attain the SDGs, which core standard emphasis "leaving no one behind", including the most marginalized groups and countries. That is, any further obstacle means the 2030 Agenda for Sustainable Development will very nearly be lost without sweeping policy responses [6]. For instance, as of August 4, 2022, the cumulative number of Sars-Cov-2 cases in WHO African Region stood at 8,749,477, with 173,254 fatalities [11]. Even though these figures may appear that Africa did largely well in containing the pandemic, the figures do not necessarily reflect the "true" picture, as low rate of reported infections is often the consequence of lack of testing capacity and a culture of under-reporting in most developing countries [2, 6]. Furthermore, given that Sars-Cov-2 ravaged even the most robust public health systems in the world such as the United States of America and United Kingdom, it would be very misleading to glorify in the low fatalities recorded in Africa [1, 5]. Unless deliberate, sustainable, and urgent interventions are made to mitigate the cataclysmic impact of the pandemic on the most marginalized groups such as PwDs, achieving the SDGs by 2030 will likely slip out of reach [6].

In Ghana, PwDs are one of the largest and deeply vulnerable groups [12, 13]. Commenting on the threat of Sars-Cov-2 on PwDs in Ghana, the executive director of the Ghana Federation of Disability (GFD) argued that most of their members lived in poor and deplorable conditions and are not able to acquire personal protective materials to protect themselves and their families from the pandemic [14]. Moreover, traditional and religious beliefs and deleterious stereotypes which often characterize interpretations of disability led to the creation and reinforcement of repressive conceptualizations of disability in the country [12, 15]. Meanwhile, the inception of the National Health Insurance Scheme in 2003 in Ghana was expected to help arrest the financial barriers to access healthcare for all citizens, most especially the most marginalized groups. Sadly, not much has changed because the exemption policy which captured vulnerable groups, such as children and the aged, conspicuously excluded PwDs [16].

Ghana has limited provisions of healthcare for PwDs despite the Disability Act, Act 2006, which aims to ensure access to effective healthcare and adequate medical rehabilitation services to members of this vulnerable group [17]. Meanwhile, PwDs are confronted with a

myriad of barriers in accessing healthcare in Ghana, including health financing, structural and physical environment [18]. For instance, Mensah et al. [17] reported that health facilities do not provide disability friendly services, making it difficult for most PwDs, especially those in wheelchairs to access hospital buildings and climb onto medical examination beds [19]. Accordingly, district accessibility audit data in Ghana, like that from Sekondi-Takoradi Metropolis (STM), revealed that 76.6% of medical centers lack internal policies targeting medical facility access for PwDs. It was further reported that 57.4% of health facilities were without accessible structures and physical environments for those who use wheelchair [20]. Moreover, evidence indicated that there are no provisions for sign language in most facilities for the needs of hearing/speech impaired patients [19]. The situation may compel PwDs to either remain at home and do nothing, self-medicate, or depend on herbal medicine, which they may abuse, making their situation devastating to health outcomes. Meanwhile, healthcare for all is a human right which must accrue to everyone irrespective of status.

As at August 5, 2021, Ghana's Sars-Cov-2 case-count stood at 168,684, with Western Region accounting for 8,647 cases, the third region with the greatest number of Sars-Cov-2 cases in Ghana [11]. Given this background, it was important to explore how the Sars-Cov-2 crisis further compounded access to healthcare among PwDs. Meanwhile, the scanty existing literature about Ghana on the subject [19, 21, 22] lack depth and also are outside the domain of Sars-Cov-2. We are of the view that the challenges and barriers to access healthcare by PwDs were compounded during the Sars-Cov-2 crisis.

## Theoretical framework

Disability is a "complex, dynamic, multidimensional and contested" construct [22]. The study of disability always benefit from the Human-Rights Model to Disability [23]. Though the human-rights model to disability does not attempt a definition for disability, it recognizes it as an "evolving concept" [23]. Therefore, PwDs are considered bonafide subjects of rights, and not mere objects of charity [24]. The promulgation and adoption of the Convention on the Rights of Persons with Disabilities (CRPD) in 2006 brought a significant shift in the promotion and protection of the rights of PwDs [23]. The aims of CRPD are to promote and ensure inclusion and full participation of PwDs in all segments of society, providing guarantees, "accountability and legal obligations on states" [25].

Within the heart of the CRPD are the ideals and codes of non-discrimination and equality of opportunities, including reverence for innate dignity and individual autonomy, non-discrimination, complete and meaningful participation and engagement in community, reverence for difference and acceptance as a share of human diversity and humanity, equality of opportunities, accessibility, equality among genders and, reverence for the growing capabilities of young people living with disability [24, 26]. Therefore, the human-right model places obligations on the state and its agencies in meeting the requirements of both the medical and social models to disability.

However, the human rights model did not predict the impact of a global crisis such as Sars-Cov-2 pandemic on access to healthcare by PwDs, especially in the African context. Unfortunately, PwDs who are already disadvantaged in accessing healthcare [19, 21, 22, 27], become worsen as a result of the Sars-Cov-2 pandemic. Meanwhile, the human rights model is valuable in expediting the researchers' appreciation of the barriers to PwDs when accessing healthcare. In addition, the model guided the researchers in analyzing the data, and developing relevant recommendations that could influence existing policies on access to healthcare by PwDs. Accordingly, the aim of this study was to explore how the Sars-Cov-2 pandemic influenced access to healthcare by PwDs in the STM.

## Methods

Leveraging on Phenomenological Analysis (PA), a thorough account of the experiences of PwDs in accessing healthcare during the Sars-Cov-2 crisis in the STM was analyzed. As a method, PA comprises revealing the intentionality and horizons of consciousness needed to unveil the world with precise meanings [28]. The objective of unveiling the intentionality of participants is the definitive purpose of participating in phenomenological research. Idhe [29] chronicled the broad attitudes the investigator must possess during data transformation. He highlights the need in addressing the experiences as articulated, describing rather than explaining the data, and horizontalizing rather than creating an importance order during data analysis. Therefore, stigma and discrimination, cost and availability of transport, poor attitude of healthcare staff, poor communication, hospital environment and equipment, handwashing and sanitizing facilities, unsuitable washrooms, cost of healthcare, registration and renewal of NHIS cards, and loss of income were considered in this study. According to the UN [30], the most prevalent disabilities in Ghana are visual impairments (40.1%), followed by physical disability (25.4%), deaf or speech impairment, and others. Hennink, Kaiser, and Marconi [31] posit that in qualitative research, saturation is normally attained by the 9th interview. Therefore, participants in this study included 17 PwDs in STM, that is GBU– 9, GSPC– 5, and GNAD– 3.

### Instrument and ethical consideration

We used a 25-item interview guide to collect data for this study. The interview protocol probed the background information of the PwDs (i.e. age, gender, level of formal education, and work experience), Sars-Cov-2 awareness and knowledge, socio-cultural factors underlining the *barriers to access to healthcare*, *disability readiness of providers*, *awareness and knowledge about Ghana's Disability Act 2006* [5, 12, 18] *Some of the open-ended questions included*: *"How is Sars-Cov-2 contracted?"*, *"Persons with disabilities are at greater risk of contracting Sars-Cov-2 than others. What are your views on this?"*, *"What difficulties did you face in getting transport to the hospital during the Sars-Cov-2 crisis?"*, *"Describe how appropriate the hospital environment is to PwDs"* and *"Describe, in general, how Sars-Cov-2 affected you while on the way to seek healthcare"*.

*The initial prepared instrument was reviewed* by the regional chairman of Ghana Federation of Disability Organization (GFDO). The instrument was then given to two physically disabled university students (in wheel chair), a Senior Lecturer in Special Education at the Department of Educational Psychology, University of Cape Coast (UCC), for further assessment.

We strictly complied with the Sars-Cov-2 safety protocols during data collection in February 2021, like socio-physical distancing, use of alcohol-base hand sanitizers and wearing of mask. We contacted the participants via their leaders and met them (PwDs) at their various meeting venues for the interviews. As the group, PwDs in STM have scheduled meeting days where they meet and discuss matters concerning the members. Therefore, we conducted the interviews with individual members on the meeting days. Though the instrument was developed in English, professional translators from the Ghanaian Language Department in UCC did translate it into preferred dialects; English, Sign language, Ewe or Fante/Twi. A professional sign language interpreter, recommended by GNAD in Takoradi, volunteered and assisted in the data collection.

To guarantee confidentiality and anonymity of participants, feedbacks were veiled rather than the use of actual personal names [5, 12]. The interviews took between 30 and 45 minutes. The interviews were audio taped alongside note taking [32]. We a psychologist on stand-by to assist any participants that may be disturbed during the interview session. The University of

Cape Coast Institutional Review Board (IRB) approved the research protocols (ID: UCCIRB/EXT/2021/06). Participants were assured of anonymity and confidentiality of their information. Moreover, they gave either verbal or written informed consent prior to participating in the study. We offered no reward to any participant, and the study was devoid of compulsion; because participants were informed, they could discontinue the study at any time without a consequence.

### Data analysis

The Phenomenological Analysis (PA) method was engaged in analyzing the data, i.e. the transcripts. The analysis involved reading the transcript several times to understand the meanings expressed by the participants, identifying significant phrases, and restating them in overall bases, which was executed by author one. These strategies aided in formulating meanings from the transcripts and certifying them through research theme discussions to reach consensus. The themes identified were further developed and organized for ease of understanding, and based on the study purpose. The second authors read the emerged themes and all other team members helped to resolve all discrepancies.

Using the PA, manual transcriptions were developed for each recorded audio and all actual personal names revealed in the interviews were anonymized using pseudonyms to guarantee confidentiality. Trustworthiness was attained by following the principles of credibility, dependability, confirmability, transferability, and authenticity [33]. Credibility through member-checking was conducted by sharing interviews and interpretations with respondents at their subsequent meetings to check for genuineness. There were additions and subtractions based on the feedback obtained from the respondents. Comprehensive notes were taken during analysis and interpretation as the study evolved to achieve confirmability of data. Finally, authenticity was reached via description of experiences of study subjects [5, 12].

## Results

We collected data from 17 participants including, GBU–nine, GSPC–five, and GNAD–three. Using the PA approach, eight themes emerged, including: awareness and knowledge, stigma and discrimination, cost and availability of transport, poor communication and attitude of healthcare staff, and hospital environment and equipment, unacceptable hygiene facilities, loss of income and cost of healthcare, and registration and renewal of national health insurance cards.

### Theme one: Awareness and knowledge

The results revealed that PwDs in STM were aware of and had a good knowledge about the cause, signs and symptoms, mode of transmission, effects, and the preventive measures of Sars-Cov-2 infection. Radio, television, and social media were the main sources of their information about the disease. However, the respondents hold some misconceptions routed in their religious believes; A visually impaired woman (VIUFP2) said:

> You see, even the Doctors and scientists don't know much about the disease, and yet it has taken over the whole world. . . What is happening now is to remind us of the role of God in the affairs of mankind. Only God can protect us.

A deaf-and-dumb (D and D) woman (DDFSAP6) said: "The disease is not common among the 'D and D' because we don't talk. It is through talking that the disease is transmitted more. This is why we're to wear nose mask." Some respondents doubt reports of people dying from

the disease in STM. According a Physically Challenged (PC) man (PCMTP4): "Because of our local foods, we're naturally protected from Sars-Cov-2. We may contract it but we would not die from it." Concerning the Ghana's Disability Law, Act 715, the respondents were not very familiar with the role of the policy in improving access to healthcare by PwDs. However, the respondents were aware that PwDs could access hospitals available to the general population.

A visually impaired (VI) female (VIUFP7) said: "I seriously do not understand how one disease could spread across the whole world. I have never experienced this in my entire life."

A visually impaired (VI) man (VIUMP8) explained:

> My mother told me about this disease (Sars-Cov-2). She explained that the disease is spread from close contact with infectious persons and so I should limit my movements. But, because I cannot see (VI), I am unable to describe the signs and symptoms.

## Theme two: Stigmatization and discrimination

The result showed that stigma and discrimination against PwDs in STM became pronounced during the first few months of the Sars-Cov-2 outbreak in Ghana. This was partly due to the belief that PwDs were potential carriers of the Sars-Cov-2 pandemic. A VI woman (VIUFP3) shared her experience:

> In our society, if you are born with a disability, then you are a punishment for the sins of your parents or family. But if you became a PWD later in life, then it is a punishment or curse for personal sins. This belief became worse since the Sars-Cov-2 outbreak in Ghana. Taxi drivers may not even want to pick you and no one wants to get close to you because they believed you could be carrying the disease. I went out one morning to buy kooko (local porridge) by the roadside but was refused. According to the seller, I was the first person to buy that morning and given my condition (PwDs), she wouldn't make good sales that whole day if she sold to me before anybody else. So, I waited for almost 20mns till someone else, a 'normal person', came to buy first before she served me.

Participant "PCMTP4" explained:

> I had a bitter experience when I joined taxi on my way to the hospital. There were two passengers already on board, one in front and the other at the back seat. As I was about joining the taxi, the passenger at the back retorted 'I don't want to contract any Sars-Cov-2', then alighted. So, I prefer to go to the drug store than go to the hospital.

Participant "DDFSAP6" also observed:

> I had a quarrel with my sister at home and she told me I was a curse on the family and an evil person. There were times when she refuses to eat with me for fear of giving birth to a "D and D" child. I sometimes fault God to have been born this way. Since the Sars-Cov-2 outbreak, my sister doesn't take me to the hospital again because she doesn't feel safe out there with me; I now go alone.

Healthcare facilities are not any safer when it comes to stigmatization and discrimination against PwDs. This was expressed by "VIUFP2":

> For those of us who are blind, it is a real nightmare to go through the hospital service successfully. On one such occasion at the hospital pharmacy, I waited for nearly 20mns without

being called so I requested to know why I was not being attended to. I became a bit impatient because I heard them called seven patients. Then one of the dispensing staff, a male voice, told me to be patient and wait for my turn. To my utter and everlasting shock and embarrassment, I overheard one other dispensing staff, a female voice, remarked to the colleague: 'The blind woman too said what? Why must she come to the hospital unaccompanied? Please, let us serve her before she starts cursing us here. It is too early, I do not want to have a problem with a blind woman.' Then I heard another patient retorted: 'Oh!' I could not eat when I got home that day. I cried almost the entire night. I felt worthless and hopeless. My only crime was that I complained about not being attended to. My son usually accompanies me to the hospital, but he had gone to school that day.

A visually impaired (VI) man (VIUMP9) said:

I realized during the period (Sars-Cov-2 outbreak) that some of my neighbors stopped associating with me. When I asked why they were distancing themselves from me, one of them said that I could be carrying Covid-19 because of my disability. Apart from my sister, I had no one to talk to.

A visually impaired (VI) man (VIUMP10) said:

This was the most terrible time in my life. My aged mother was the only one I could communicate with. No one would talk to me. I became a bit depressed and suicidal at a point. God bless my mother for her support and care around that time (Sars-Cov-2 outbreak).

## Theme three: Cost and availability of transport

Cost and availability of transport also emerged as a key barrier to access healthcare by PwDs in STM, especially during the Sars-Cov-2 outbreak. The results showed that most PwDs in STM could not afford the cost of transport to the hospital especially during the initial stage of the pandemic. Respondents relied on their relatives and friends for transportation and support to the hospital. Participant "VIUFP2" shared her experience:

Transportation to the hospital has always been a challenge, even before the Sars-Cov-2 outbreak. My sister pays for my transport to the hospital and whenever she doesn't have money, I stayed home. Because I could not go to the hospital unaccompanied, my sister had to pay double fare each time I went to the hospital."

A PC woman (PCFTP5) explained:

I was always charged double the fare each time I pick a taxi to the hospital. I don't really blame the drivers because some passengers may either alight or not join when they see me on board. It is one concern that affects most PwDs, especially the PC and the blind. This actually became worse since the onset of Sars-Cov-2 in Ghana.

A visually impaired (VI) woman (VIUFP11) explained:

Because of my situation (VI) I could not go anywhere without assistance. My sister pays for my transport and had to accompany me wherever I went. It was difficult getting vehicle sometimes because some drivers may not want to pick me.

A visually impaired (VI) woman (VIUFP12) said:

I have a niece who accompanies me to town, including hospital. It was very difficult getting vehicle to the hospital during the period (Sar-Cov-2 outbreak) as most drivers were either not prepared to pick me or would normally charge me double fares.

## Theme four: Poor communication and attitude of healthcare staff

Though these are age-old barriers, they became worsened during the Sars-Cov-2 outbreak. Participant "VIUFP2" shared her frustration:

We are always spoken to harshly by the hospital staff. You could be waiting either at the OPD, consulting room, or even the dispensary among other patients for a long time. The staff of the hospital do not realize early that we are blind until after shouting at us. These nurses shout at you as if you are just wasting their time. There is just no provision for PwDs, not even in this period of Sars-Cov-2.

Again, participant "PCFTP5" said:

As for me, even though the nurses are generally rude towards us, the doctors are not any better. After waiting for nearly two hours in the queue to see the doctor, I finally entered and noticed the doctor was sending an email. I sat there for 10 minutes before he attended to me. He told someone on phone that he has sent the email so, the person should check and get back to him. He didn't even excuse me nor said sorry for keeping me waiting.

Sharing his experience, participant "PCMTP4" said:

Nurses are the guiltiest of poor staff attitude towards us. I seriously don't know why they behave so rudely towards us. Some do not even want to touch us because they think we are dirty and could be carrying Sars-Cov-2 virus. These nurses are just something else. Facebook and social media are all they know, all their attention is on the phone and so if you want their assistance a bit, they become irritated and rude to you.

Participant "VIUFP2" again:

For those of us who have no caretaker to accompany us to the hospital, we go through hell in the hands of these nurses. The hospital doesn't have anyone to assist you through the process. If you are lucky to have a compassionate patient, then he or she would assist you through the processes. Now, because of Sars-Cov-2, nobody will even mind you.

A PC woman (PCFTP13) exclaimed: ". . .Look! Long before this Covid-19 disease, the nurses have always been very rude to us (PC). Yes, I was at the hospital around that time (Covid-19 outbreak) and the hospital staff would not even talk to you."

A PC man (PCMTP14) said: ". . .As I entered the consulting room, I was made to sit far away from the nurse and doctor, and I could not hear them well because they were in nose mask."

Poor communication become a barrier to access to healthcare by PwDs in STM during the period of Sars-Cov-2 crisis. A participant "VIUFP3" retorted:

Hm!!! How do I even say this? My problem is even more about the pharmacy staff. Instead of taking time to explain to us how the medications should be taken, they just hand them

over to us. I once told them to exercise patience while explaining how to take the medica-tions, but the staff shouted at me saying: 'but you can't see, so how else do I explain to you?' I felt very humiliated by her reaction, but what could I do? In my view, they disrespect PwDs because they have not been trained in how to take care of us, especially those of us who are blind.

Participant "DDFSAP6" said:

I was thinking that with Sars-Cov-2 around, hospitals would have looked for sign language interpreters to explain the hospital procedures to us. I couldn't even touch anyone for assis-tance because of the fear of Sars-Cov-2 humiliation. Sometimes you can tell the staff are insulting you but because I could not hear I just endure it.

A D and D woman (DDFSAP16) said: "I went to the hospital alone and there was no one to assist me through the process. In fact, I could not tell how the Doctor interpreted my condition."

## Theme five: Hospital environment and equipment

The results also showed that most hospital environments were not disability friendly. Narrat-ing his experience, participant "PCMTP4" explained:

Almost all the hospitals I visited in STM do not have adequate structures to support PwDs. Though a few of the hospitals have pavements for wheelchairs, they were too steep. The floor tiles used are also too smooth. So, what is the essence of the Disability Act if our hospi-tals still discriminate against PwDs? Anyway, these days because of Sars-Cov-2, unless I'm very ill, I don't go to the hospital.

Participant "PCFTP5" exclaimed:

Just look at the location of the Regional Hospital, it is sited on a hill and you can't use the wheelchair. This is why I don't like going there, it is a last resort. Unfortunately, these are government hospitals and you would expect them to comply with state laws (the Disability Act) but that is not the case.

A PC woman (PCFTP15) explained:

. . .The hospital environment was not disability friendly at all because most of their service points were inaccessible to those of us in wheel chair. But for the assistance of one patient, I could not have accessed the pharmacy for my drugs.

## Theme six: Unacceptable hygiene facilities

The findings indicated that the available hygiene facilities (handwashing, sanitizing and wash-rooms) are unacceptable for use by PwDs during the Sars-Cov-2 crisis. According to partici-pant "PCMTP4":

As someone in wheelchair, it was very difficult to use the Veronica Buckets provided at the hospitals. Even the hand sanitizers they provided were placed on shelves and tables that are not accessible to those of us who use wheelchairs.

Another VI man (VIUMP1) said:

The hospital doesn't care that some of us are blind and would need assistance in using these facilities. Sometimes you wish to wash your hands but there is no one to assist you. As for me, because of Sars-Cov-2, I always have my own hand sanitizer in the pocket because I know I couldn't use what was at the hospital.

Respondents were sad about the lack of washroom facilities suitable for all disability groups. Participant "PCMTP4" said:

It has always been my prayer never to feel the need to use the washrooms at the hospitals. Those of us with disabilities are already at risk yet the hospitals don't care. All they know is money, money, money. The washrooms are smelly and not even clean, something must be done about this.

Participant "DDFSAP6" also lamented:

The washrooms in most hospitals are not well kept. I used the washroom of one hospital in June, 2020, and I regretted. There was no toilet roll and the cistern was also leaking with water on the floor. How do you prevent Sars-Cov-2 when hospital washrooms were not clean? There was no soap to even wash your hand after using the washroom.

Participant "VIUFP7" said: "Well, I cannot describe how neat the place was because I cannot see (VI) but the smell was very bad."

Participant "VIUMP8" explained: "What? Hospital washrooms? I make sure I do not use it at all. Thank God I have never had the need to."

## Theme seven: Loss of income and cost of healthcare

The result showed that some PwDs and their caretakers lost their jobs during the Sars-Cov-2 outbreak which further aggravated their inability to pay for high cost of healthcare. Sharing her experience, participant "PCFTP5" said:

I sell fried fish but since this disease (Sars-Cov-2) started, I couldn't go to the market to sell, and my sister who supports me also lost her job as a private school teacher. Life has become very difficult for me. Sars-Cov-2 is an evil disease and we need God's intervention before it would go." Participant "DDFSAP6" also said: "I was a shop attendant and earn a monthly allowance of 200.00 cedis, but since the outbreak of Sars-Cov-2, my boss asked me to stay home until further notice.

Moreover, some hospitals charge PwDs, for services covered under NHIS. Also, some hospital dispensaries lack essential NHIS medicines so patients had to buy these medications from private pharmacies. Participant "VIUMP1" explained, sadly:

You see, I have diabetes and hypertension and yet I don't always get my prescribed drugs at the hospital. Just imagine in my condition, and the fact that I no longer work. How do I get money to buy these drugs? I really don't see the importance of the disability act because–it doesn't help me in anyway.

Participant "VIUFP3" said: "Even with the NHIS card, you would still buy drugs, so what does that mean? Something must be done to make sure that PwDs have access to free medical care."

Participant "PCFTP13" explained:

. . .They gave me only two out of the six drugs prescribed and asked me to buy the rest in town. Since I do not have money to buy the rest, I just used what they provided. I do not know what the NHIS covers.

### Theme eight: Registration and renewal of national health insurance cards

The result revealed that most PwDs find it rather difficult to register and renew their NHIS cards. Narrating her experience, participant "DDFSAP6" said: "The last time I visited the NHIS office to renew my card, I was asked to go for an introductory letter from the Department of Social Welfare, confirming my disability status before I would be attended to." Another PwDs, participant "PCMTP4" explained:

I have been to the NHIS office on two different occasions and was told the network was down so I couldn't register. Most PwDs are unable to register and renew their NHIS cards because of their inability to pay the premium. Long queues at the NHIS offices also discourage us from going there.

Participant "PCMTP14" said:

. . .After managing to get to the NHIS office, they told me to get clearance from the Social Welfare office before they could renew my card for free. Considering the cost of transport to the Social Welfare office, I had to just pay for the renewal by myself.

## Discussion

We explore the impact of Sars-Cov-2 pandemic on access to healthcare by PwDs in the STM. The Sars-Cov-2 pandemic has seriously exposed major setbacks in the implementation of Ghana's Disability Law, Act 715, in relation to access to healthcare by PwDs. Nonetheless, PwDs in STM were knowledgeable about the Sars-Cov-2 pandemic and understand that hospitals available to the general population are accessible to them (PwDs) as well. Though PwDs are always faced with healthcare access barriers, it appears the situation is compounded in the wake of the Sars-Cov-2 pandemic, especially in areas like stigmatization and discrimination, cost and availability of transport, poor communication and attitude of healthcare staff, hospital environment and equipment, unacceptable hygiene, lost of income and cost of healthcare, and registration and renewal of NHIS cards. The Sars-Cov-2 pandemic is therefore disproportionately impacting PwDs than the general population.

Contrary to widely published suggestions that PwDs are less likely to have access to information regarding the Sars-Cov-2 pandemic [4, 5, 34], our finding revealed that PwDs in STM were aware and knowledgeable about the disease, its causes, mode of transmission, signs and symptoms, and the preventive measures. This could be credited to the aggressive disability friendly educative campaigns about the diseases on television, radio, social media, and more importantly at their meetings. Our analysis revealed that extensive education concerning the pandemic had gone at the meetings of PwDs in STM.

Even though a study from sub-Saharan Africa suggested that PwDs were not aware that hospital available for the general population could be accessed by them (PwDs) [35], our finding revealed otherwise. Perhaps, there is an urgent need to restructure existing healthcare facilities to effectively accommodate and care for PwDs. Consistent with a study from Tamale Metropolis [16], PwDs at STM lack knowledge about Ghana's Disability Law, Act 715,

therefore, unable to demand those facilities when accessing healthcare. In the face of inability to demand for their rights and privileges, healthcare organizations and providers may continue to denial PwDs proper access to healthcare.

We again found a sustained stigmatization and discrimination against PwDs in STM especially in the first few months of the Sars-Cov-2 outbreak. Contrary to the provisions of the Ghana's Disability Law and the United Nations Human Rights Office guidelines on Sars-Cov-2 and the rights of PwDs [36, 37], the findings revealed gaps in medical care for PwDs in STM. The healthcare providers, nurses and doctors, were hesitant in touching PwDs during examination, a situation which creates psycho-emotional consequences for the patient. Unfortunately, PwDs suffered heightened levels of marginalization and social exclusion during the global crisis. PwDs find it difficult to pick transport, go out to buy food, seek healthcare, and interact with relatives and friends without experiencing discrimination. This situation might have increased their exposure of PwDs to the risk of Sars-Cov-2.

The findings further disclosed that transport cost and availability remain a relevant barrier to access to healthcare by PwDs, which intensified during the Sars-Cov-2 crisis. For instance, some transport owners charged PwDs double fares for occupying double seats, thus, making healthcare access difficult. In some cases, it is the support from relatives, friends, and kind-hearted members of the community that aid PwDs access healthcare. Research from Namibia [38], Sudan and Malawi [38], South Africa [39], and Ghana [18, 40] revealed similar situations. More recently, Baart and Taaka [35] argued that PwDs in developing countries had to contend with not only the cost of transport for themselves, but the transport fare and financial incentive of those accompanying them (PwDs). This compounds access to quality healthcare resulting in delayed healthcare seeking and attendant poor health outcomes.

Poor attitude of healthcare staff is another existential barrier to access healthcare by PwDs in STM. Though this has always been a major concern for PwDs, it became worse during Sars-Cov-2 crisis. Apart from Ormsby et al. [41] reporting otherwise, poor attitude of healthcare staff was widely reported as a serious drawback to access to healthcare by PwDs, especially in developing countries [42–47]. Persons with disabilities had to endure regular bouts of verbal and psychological abuse, and neglect whenever they visited the hospital. Sars-Cov-2 increased poor staff attitude which resulted in long waiting time for crucial healthcare services needed by PwDs, the quality of care is thus, compromised. Unless they are critically ill and needing emergency care, PwDs take to self-medication and use of herbal concoctions to deal with their ailments during the Sars-Cov-2 crisis. Previous findings also reported that healthcare staff appear insensitive, whether wittingly or due to inadequate training, to the rights and needs of PwDs [19, 35, 42, 48, 49]. Moreover, evidence [19, 48] argued that the poor staff attitudes sustained against PwDs over the years, creates a sub-culture of stigma, discrimination, and neglect. Unfortunately, in such environment, PwDs are more likely to stay away from healthcare facilities even when they are ill, and depend on self-medication and unorthodox herbal drugs.

Such poor attitude manifests also poor communication between healthcare providers and PwDs during the Sars-Cov-2 crisis. Contrary to Sars-Cov-2 guidelines and recommendations [34, 37, 50] and Ghana's Disability Law, we observed that information about Sars-Cov-2, general hospital procedures and processes, and hospital services were not provided in multiple accessible formats to reach all PwDs. For example, there was very little provision for sign language, augmentative and alternative communication for PwDs during the Sars-Cov-2 crisis. This could result in wrong diagnoses and prescriptions, wrong medications usage, unclear instructions about how to take medications, long delays in receiving care, lack of patient participation in the care process, lack of privacy and confidentiality, and adverse health outcomes for PwDs.

The health facilities in STM have little or no provision for use of wheelchairs and clutches, floor tiles are said to be too smooth to support clutches, examination beds are too high, narrow

doors, lifts to storey buildings have no provision for the PC, and most of facilities are located on hills. The hospitals with pavements for wheelchairs were reported to be too steep and without ramps. The situation mirrors previous studies [35, 42, 43, 48, 51–53] suggesting inaccessible health facilities and equipment by PwDs in developing countries. Moreover, Gaihre [42] revealed that the PC and VI were the most impacted by the barrier of physical access to healthcare facilities, a condition which makes it difficult for such people to access healthcare during the Sars-Cov-2 crisis.

Handwashing and sanitizing facilities were also found to be unsuitable for PwDs, exposing them to Sars-Cov-2. Hand-washing and sanitizing facilities though found to be effective in the prevention of Sars-Cov-2 [34, 50, 54], PwDs in STM could not utilize these facilities provided by hospitals because they were unsuitable. Thus, disability dynamics required for setting up these Sars-Cov-2 safety facilities were ignored in clear breach of UN conventions on disability [1, 34]. Given this gap, PwDs in STM were hesitant in accessing healthcare during the Sars-Cov-2 crisis. Moreover, hospital washrooms were identified as very unsuitable for use by PwDs, especially during this time of Sars-Cov-2 crisis. Basheer [5] argued that washrooms have no toilet rolls, irregular flow of water, most were smelly and unclean, and do not have soaps for washing of hands. Considering the infectious nature of Sars-Cov-2, hospital washrooms are required to be suitable for all categories of patients, especially for PwDs, so as to prevent or minimize the infection rate [34].

Cost of healthcare was another barrier to access healthcare by PwDs in STM. Baart and Taaka [35] have found that PwDs do not have sustainable sources of income and yet have higher healthcare needs due to their impairment, and are likely to incur higher cost of medical care. To mitigate the financial burden of healthcare for PwDs in Ghana, Ghana's National Health Insurance Policy 2003 provided a special dispensation for PwDs to register and renew their NHIS cards for free [55, 56]. However, some healthcare facilities are said to charge PwDs for services, even those covered under the insurance scheme. Meanwhile, some of these PwDs have lost their jobs during the Sars-Cov-2 outbreak and have no means of sustenance except for support from relatives. As argued by Abrokwah [18], while the high cost of healthcare has a negative impact on access to quality care for everyone, PWDs are three times more likely to report unmet health needs due to cost [57].

The cost of transport to NHIS offices, network problems at the NHIS registration centers, insistence on introductory letters from the Social Welfare Department before enjoying free registration and renewal of cards, fear of contracting Sars-Cov-2, and lack of disability friendly washrooms and wheelchairs at NHIS registration centers for PwDs, contributed to the inability of PwDs in STM to register and renew their NHIS card to enable them seek healthcare. Unfortunately, many of the PwDs and their family relatives have lost their sources of income because of Sars-Cov-2 [7]. Some who worked as shop attendants were laid off as a result of poor sales, and those into trading experienced low patronage, especially during the periods of lockdown. Imoro [16] also indicated that the loss of income negatively affected the lives of most PwDs and prevented them from accessing healthcare during the Sars-Cov-2 crisis. Therefore, until pragmatic and proactive efforts are made, PwDs will continue to have restricted access to quality healthcare especially during infection disease emergency.

## Conclusion

We found that PwDs in STM experienced elevated levels of barriers to access to healthcare since the outbreak of Sars-Cov-2 in Ghana. These barriers encountered along the pathway to care, include stigmatization and discrimination, cost and availability of transport, poor communication and attitude of healthcare staff, hospital environment and equipment,

unacceptable hygiene facilities, loss of income and cost of healthcare, and registration and renewal of NHIS cards. The Sars-Cov-2 pandemic is therefore, disproportionately impacting PwDs and exposing the gaps in the implementation of Ghana's Disability Law, Act 715. Although the United Nations Sustainable Development Goals (SDG) 11.2 called for the delivery of access to safe, cost-effective, accessible and sustainable transport regimes for all persons, particularly the vulnerable like PwDs, access to quality healthcare services remains a significant challenge for PwDs. Therefore, the need to identify and eliminate the access barriers to quality healthcare services for PwDs, especially during periods of global crisis such as the Sars-Cov-2 pandemic, is obvious imperative. Findings of the study re-focuses the attention of hospital managers in STM to the healthcare needs of PwDs in STM, and provide a suitable context for potential policy design and analysis in the area of disability rights.

## Limitations

A sample of 17 participants limited the generalizability of the research findings. Again, some respondents were a bit emotional in their responses to some questions, especially regarding attitude of healthcare staff. However, the results give a good account of the lived experiences of PwDs in STM in accessing healthcare during the Sars-Cov-2 crisis, and could provide a basis for future studies on the disability friendliness of healthcare facilities in Ghana.

## Acknowledgments

We are grateful to all PwDs in STM for sharing their experiences with us. Again, we expressed our profound appreciation to the Presidents of the various disability groupings in STM for granting us permission to conduct the study. We are equally grateful to our Sign Language interpreter, Mr Isaa Abakah, and Flight Lieutenant Lucy Adjanor Akoto for proof reading the final work.

## Author Contributions

**Conceptualization:** Nkosi Nkosi Botha, Fortune Selase Atsu, Edward Wilson Ansah.

**Data curation:** Nkosi Nkosi Botha, Fortune Selase Atsu, Edward Wilson Ansah, Cynthia Esinam Segbedzi, Sarah Annim.

**Formal analysis:** Nkosi Nkosi Botha, Fortune Selase Atsu, Edward Wilson Ansah.

**Investigation:** Nkosi Nkosi Botha, Cynthia Esinam Segbedzi, Sarah Annim.

**Methodology:** Nkosi Nkosi Botha, Edward Wilson Ansah.

**Supervision:** Edward Wilson Ansah.

**Validation:** Nkosi Nkosi Botha, Cynthia Esinam Segbedzi, Sarah Annim.

**Writing – original draft:** Nkosi Nkosi Botha, Fortune Selase Atsu, Edward Wilson Ansah, Cynthia Esinam Segbedzi, Sarah Annim.

**Writing – review & editing:** Nkosi Nkosi Botha, Fortune Selase Atsu, Edward Wilson Ansah, Cynthia Esinam Segbedzi, Sarah Annim.

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
