## [Decision Letter · Decision Letter 0]

3 Feb 2023

PGPH-D-22-01986

Impact of COVID-19 on access to healthcare by persons with disabilities

Dear Dr. BOTHA,

Thank you for submitting your manuscript to PLOS Global Public Health. After careful consideration, we feel that it has merit but does not fully meet PLOS Global Public Health’s publication criteria as it currently stands. Therefore, we invite you to submit a revised version of the manuscript that addresses the points raised during the review process.

We look forward to receiving your revised manuscript.

Kind regards,

Joao Tiago da Silva Botelho

Academic Editor

Journal Requirements:

Additional Editor Comments (if provided):

Reviewers' comments:

Reviewer's Responses to Questions

**Comments to the Author**

1. Does this manuscript meet PLOS Global Public Health’s publication criteria? Is the manuscript technically sound, and do the data support the conclusions? The manuscript must describe methodologically and ethically rigorous research with conclusions that are appropriately drawn based on the data presented.

Reviewer #1: Yes

Reviewer #2: Partly

2. Has the statistical analysis been performed appropriately and rigorously?

Reviewer #1: N/A

Reviewer #2: No

3. Have the authors made all data underlying the findings in their manuscript fully available (please refer to the Data Availability Statement at the start of the manuscript PDF file)?

Reviewer #1: Yes

Reviewer #2: No

4. Is the manuscript presented in an intelligible fashion and written in standard English?

Reviewer #1: Yes

Reviewer #2: No

5. Review Comments to the Author

Reviewer #1: Manuscript number: PGPH-D-22-01986

Article Type: Research Article Article Type: Research Article

Full Title: Impact of COVID 19 on access to healthcare by persons with disabilities

Short Title: COVID 19 and persons with disabilities

Corresponding Author: Nkosi Nkosi BOTHA

University of Cape Coast

Cape Coast: GHANA

Order of Authors: Nkosi Nkosi BOTHA

Fortune Selasi ATSU, MPhil Candidate

Edward Wilson ANSAH, PhD

Cynthia Esinam SEGBEDZI, PhD Candidate

Sarah ANNIM, MPhil Candidate

I have gone through the paper and I found that it is of great public importance. It’s explicitly unrevealed concern of people living with different kind of disabilities in Africa context and possible effect to access o health care in unprecedented health emergencies, particularly COVID-19 in this context. It is important that data collection tool was given to scholars with disability for evaluation. However, it was not clearly described why was it conducted to scholars at university level but study was conducted in participant from general population group.

Here are the minor comments in some sections:

In introduction section author mentioned that “The plights of PwDs are vast and deep, for example, PwDs have often been sidelined from health information, and mainstream health provision, including access to goods and services”. It was not explicitly indicated if it is expert opinion or a finding from previous studies of which could be referenced.

Methodology

1. Methodology; well presented. Nevertheless, apart from loss of income that was tremendously noted during COVID-19 pandemic, how did you able to describe difference existing/existed between pre-COVID-19 and during COVID-19 era (what are the baseline? And reference) for the following mentioned factors:-

• Stigma and discrimination.

• Cost and availability of transport

• Hospital environment and equipment

• Poor attitude of health staff

• Poor communication

• Handwashing and sanitizing facilities

• Unsuitable washroom

• Cost of health care

• Poor communication

• Registration and renewal of NHIS card

2. Kindly describe how did you managed to collect data from “deaf and dumb”?.

Discussion and conclusion.

Interpretations are matching with results hence well presented. Author has well presented and meaning are these findings in Ghana and African context regarding access of health care in people with disabilities in during COVID-19. As commented before, there is paucity of reference on how the condition was before the pandemic.

Limitation

Samples size has been presented as only limitation.

• How about socioeconomic status; they have nothing to do with participant responses? (Or challenges they encountered during COVID-19 pandemic?).

Reviewer #2: In my opinion, I have a lot of problems with the sample size of the article. It will be difficult to popularize the results found. Why not increase the sample size before wanting to publish?

Furthermore, I would like to know the added value of the study. Everything that is put in the article, we already know. People with disabilities have always been rejected in our communities. What innovative interventions would you recommend to correct this?

So I'm waiting for those answers before I decide.

Thank you!

6. PLOS authors have the option to publish the peer review history of their article (what does this mean?). If published, this will include your full peer review and any attached files.

**Do you want your identity to be public for this peer review?** For information about this choice, including consent withdrawal, please see our Privacy Policy.

Reviewer #1: **Yes: **Ombeni Eliud Chimbe

Reviewer #2: **Yes: **DEGNONVI Horace

---

## [Decision Letter · Decision Letter 1]

3 Mar 2023

Impact of COVID-19 on access to healthcare by persons with disabilities

PGPH-D-22-01986R1

Dear Mr BOTHA,

We are pleased to inform you that your manuscript 'Impact of COVID-19 on access to healthcare by persons with disabilities' has been provisionally accepted for publication in PLOS Global Public Health.

Best regards,

Joao Tiago da Silva Botelho

Academic Editor

Reviewer Comments (if any, and for reference):

Reviewer's Responses to Questions

**Comments to the Author**

1. If the authors have adequately addressed your comments raised in a previous round of review and you feel that this manuscript is now acceptable for publication, you may indicate that here to bypass the “Comments to the Author” section, enter your conflict of interest statement in the “Confidential to Editor” section, and submit your "Accept" recommendation.

Reviewer #1: All comments have been addressed

2. Does this manuscript meet PLOS Global Public Health’s publication criteria? Is the manuscript technically sound, and do the data support the conclusions? The manuscript must describe methodologically and ethically rigorous research with conclusions that are appropriately drawn based on the data presented.

Reviewer #1: Yes

3. Has the statistical analysis been performed appropriately and rigorously?

Reviewer #1: N/A

4. Have the authors made all data underlying the findings in their manuscript fully available (please refer to the Data Availability Statement at the start of the manuscript PDF file)?

Reviewer #1: Yes

5. Is the manuscript presented in an intelligible fashion and written in standard English?

Reviewer #1: Yes

6. Review Comments to the Author

Reviewer #1: All raised queries/concern have been well addressed.

7. PLOS authors have the option to publish the peer review history of their article (what does this mean?). If published, this will include your full peer review and any attached files.

**Do you want your identity to be public for this peer review?** For information about this choice, including consent withdrawal, please see our Privacy Policy.

Reviewer #1: **Yes: **Ombeni Eliu Chimbe
